# Exploring oral microbiome in oral squamous cell carcinoma across environment-associated sample types

Zizheng Wang,[1,2,3] Yilong Chen,[1,2] Haoning Li,[4] Yuan Yue,[4,5] Haopeng Yu[1,2]

**ABSTRACT** The relationship between the oral microbiome and oral squamous cell carcinoma (OSCC) has been extensively investigated. Nonetheless, most previous studies were single-center, resulting in the absence of systematic evaluations. To address this gap, we performed a comprehensive meta-analysis on 1,255 samples from OSCC-related 16S rRNA gene data sets, representing a diverse range of OSCC phenotypes. It is recognized that the progression of cancer is related to the alterations in the microbiome among different phenotypes. Our findings revealed distinct microbiome characteristics among different sample types, with Biopsy (Bios) and Swab samples exhibiting significant differences between phenotypes. In Bios samples, the microbiomes of the Cancer group and the normal tissue adjacent to the tumor (NAT) group display a higher similarity, while both differ from the microbiome of the Fibroepithelial polyp (FEP) group. Moreover, the identified differential genera and pathways corresponded with these observations. We developed a diagnostic model using the random forest algorithm on Swab samples, achieving an area under the receiver operating characteristic curve (AUC) of 0.918. Importantly, this model exhibited considerable effectiveness (AUC = 0.849) when applied to another sequencing platform. Taken together, our study provides a comprehensive overview of the oral microbiome during various OSCC progression stages, potentially enhancing early detection and treatment.

**IMPORTANCE** This study answers key questions regarding the universal microbial characteristics and comprehensive oral microbiome dynamics during oral squamous cell carcinoma (OSCC) progression. By integrating multiple data sets, we examine the following critical aspects: (1) Do different sample types harbor distinct microbial communities within the oral cavity? (2) Which sample types offer greater potential for investigating OSCC progression? (3) How are the oral microbiomes of the Cancer group, normal tissue adjacent to the tumor group, and Fibroepithelial polyp group related, and what is their potential association with OSCC development? (4) Can a diagnostic model based on microbial signatures effectively distinguish between Cancer and Health groups using Swab samples?

**KEYWORDS** oral microbiome, meta-analysis, oral squamous cell carcinoma, 16S rRNA, oncogenesis

**Peer Reviewers** Georg Conrads, Division of Oral Microbiology and Immunology, Aachen, Germany; Pierre Le Bars, University of Nantes, Nantes cedex, France

Address correspondence to Haopeng Yu, yuhaopeng@wchscu.cn, or Yuan Yue, hxkqyueyuan@163.com.

Zizheng Wang and Yilong Chen contributed equally to this article. The author order was determined based on the length of time each author dedicated to the research.

The authors declare no conflict of interest.

Oral squamous cell carcinoma (OSCC), a subtype of head and neck squamous cell carcinoma (HNSCC), predominantly affects the oral and maxillofacial region and is characterized by a high tendency for recurrence and metastasis (1, 2). Based on the data from 2020, there was a global incidence of 377,713 new cases in oral cancer (mainly OSCC), resulting in 177,757 deaths (3). Tobacco use and/or heavy alcohol consumption have been shown to increase the risk of OSCC (4–7). Other possible risk factors include viral infections, poor oral hygiene, and *Candida* species infection (8). Additionally, certain forms of leukoplakia have long been recognized as independent risk factors for

carcinoma development (9). Dysbiosis of the oral microbiome has also been implicated in OSCC progression (10). Several hypotheses have been proposed to elucidate the role of oral microbiota in OSCC (5, 11–13). These hypotheses suggest that dysregulation in the oral microbiome may contribute to the development of OSCC through mechanisms involving inflammatory responses, cellular invasion, immune modulation, and toxic metabolite production (14, 15).

Nonetheless, previous studies have failed to consistently identify OSCC-associated microbial features. This inconsistency may arise from the focus on specific populations and geographic regions, offering only a limited perspective on OSCC ecological imbalances. Additionally, methodological differences in sample size and data processing have impeded the generalizability of these findings (16, 17). The relationship between the progression of cancer and the alterations in the microbiome among different phenotypes is acknowledged (5, 10, 12, 13). Recent evidence indicates that distinct sample types and phenotypes harbor unique microbial communities (18–20). However, persisting inconsistencies in selections of sample type and phenotype hinder meaningful comparisons between studies (17, 21). Given these limitations, we underscore the importance of a systematic evaluation and synthesis of microbial communities during OSCC progression using standardized, large-scale approaches (16).

Meta-analysis, a large-scale, cross-cohort research method, allows for the integration and analysis of an extensive volume of raw sequencing data from public data sets (22, 23). This approach mitigates the impact of biological and technical confounding factors, enabling the identification of shared and stable microbial patterns across multiple

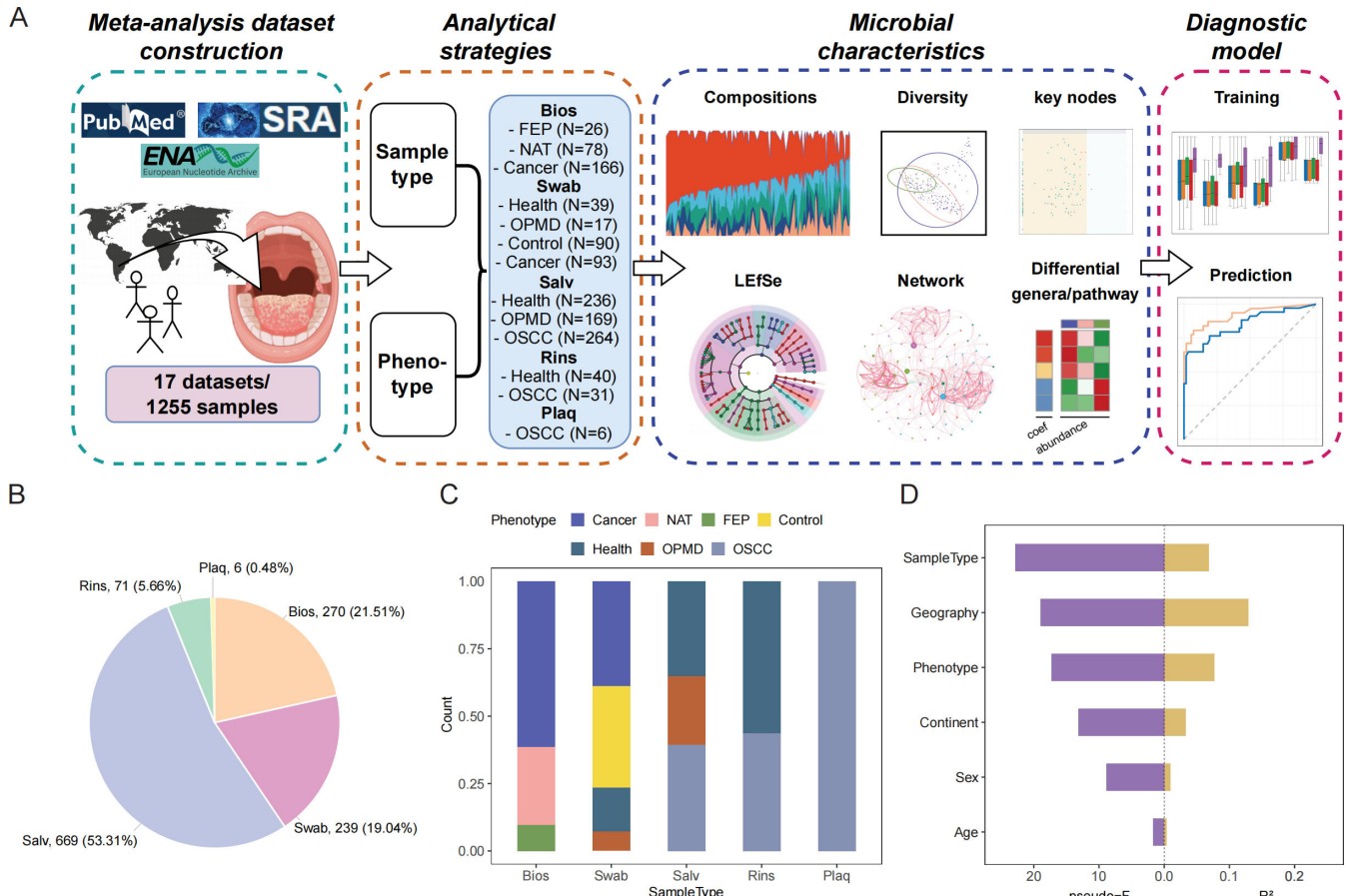

**FIG 1** Overview of OSCC oral microbiome (A) Meta-analysis strategy and workflow. (B) Distribution of sample types by quantity. (C) Proportions of different phenotypes within each sample type. (D) Bar plot illustrating the effect size (Adonis $R^2$ and pseudo–F) of categories significantly associated with oral microbial variations ($P < 0.05$, PERMANOVA). Abbreviations: PERMANOVA, permutational multivariate analysis of variance.

studies (24). In the absence of large-scale cross-cohort research on the oral microbiome in OSCC, meta-analysis can yield insights comparable to those of multicenter studies with sizable samples (25). In our investigation, we incorporated all available OSCC-related 16S rRNA gene data sets and re-analyzed the 1,255 collected samples using a standardized pipeline. First, we appraised and refined previously established analytical strategies, examining not only the characteristics of the oral microbiome across various sample types but also identifying the most suitable sample types for investigating OSCC progression. Subsequently, we conducted a systematic assessment of associations between the microbiome of different phenotypes to better understand the dynamic changes in the oral microbiome during OSCC progression. Building on these findings, we pinpointed differential microbes and predicted pathways across diverse phenotypes. Finally, we developed a diagnostic model to facilitate the early detection of OSCC.

## RESULTS

### Characteristics of the meta-analysis data set

To comprehensively understand the oral microbial characteristics in OSCC, we sought to acquire as many 16S rRNA gene amplicon sequencing data as possible (Fig. 1A). We searched the PubMed database using the keyword combinations "(('oral squamous cell carcinoma') OR ('OSCC')) AND ((('16S') OR ('metagenome')) OR ('microbiome'))," as well as the Sequence Read Archive (SRA) and European Nucleotide Archive (ENA) databases with the term "oral squamous cell carcinoma." In total, we gathered 1,255 OSCC-related samples from 17 studies spanning 13 different countries/regions. The number of cases and controls in each study was generally balanced and the study subjects' mean age was 53.13, with a male-to-female ratio of 3.36 (Table 1).

In the meta-analysis data set, each sample contained information from 17 metadata categories, including "Study Name," "Geography," "Sample Type," and "Habitat" ( Data S1 [supplemental materials can be found at DOI: 10.6084/m9.figshare.28227341]). Considering that the sample type is a main focus in clinical practice (17), we classified all samples into five groups according to the "Sample Type" category: Biopsy (Bios), Swab, Oral Rinse (Rins), Dental Plaque (Plaq), and Saliva (Salv). In relation to OSCC progression, we further dissected each group within the "Sample Type" category. These classifications were organized under the category "Phenotype." First, Bios samples were divided into three groups: FEP (Fibroepithelial polyp), NAT (normal tissue adjacent to the tumor), and Cancer. Specifically, the FEP group acts as a control setting for supplementary histological examination, mainly applicable when biopsies cannot be obtained from normal tissues (26). For Bios samples from OSCC patients, the NAT group represents normal tissues adjacent to the tumor, the most common control setting in prior studies, whereas the Cancer group represents cancerous tissues. Second, Swab samples were subdivided into four groups: Health, OPMD (oral potentially malignant disorders), Control, and Cancer. The Health group samples originate from normal tissue surfaces in the oral cavities of healthy study subjects, the OPMD group is sampled from premalignant lesion surfaces (27), and the Cancer and Control groups are sampled from cancer tissue surfaces of OSCC patients and their anatomically matched contralateral normal mucosa, respectively. Finally, Salv, Rins, and Plaq samples were separated into Health, OPMD, and OSCC groups based on the subjects' disease states, respectively (Fig. 1B and C).

### "Sample Type" and "Phenotype" categories are major focuses in this study

Considering that the hypervariable regions can influence sequencing data (22), we logically assimilated all sequencing data by mapping all reads to the full-length 16S rRNA gene sequences in Greengenes before conducting further analyses. Prior studies indicated that sample types, environmental factors, cancer progression, and others have the potential to modify the microbial composition (18, 28–30). After applying two rigorous criteria—biological relevance and an annotation proportion exceeding 70%—

**TABLE 1** List of 16S rRNA gene data sets included in the meta-analysis[a]

| Name | Year | Sample type (N[b]) | Male/female (n[c]) | Age[d] (n[c]) | Geography | Continent | Group (n[c]) | p-value ($\chi^2$) |
|---|---|---|---|---|---|---|---|---|
| Al-Hebshi | 2017 | Bios; Swab (40) | 20/20 (40) | 53.08 ± 9.7(40) | Yemen; Saudi Arabia | Asia | Bios_Cancer (20) Swab_Health (20) | 1 |
| Chang | 2019 | Bios; Plaq (17) | NA (17) | NA (17) | Shenyang | Asia | Bios_Cancer (5) Bios_NAT (6) Plaq_OSCC (6) | 0.9429 |
| Perera | 2018 | Bios (51) | Male (43); NA (8) | NA (51) | Sri Lanka | Asia | Bios_Cancer (25) Bios_FEP (26) | 0.8886 |
| Sarkar | 2021 | Bios (100) | 64/36 (100) | 52.68 ± 11.33 (100) | Kolkata | Asia | Bios_Cancer (50) Bios_NAT (50) | 1 |
| SRP358074 | NA | Bios (40) | NA (40) | NA (40) | Shanghai | Asia | Bios_Cancer (40) | |
| Yang | 2021 | Bios; Salv (76) | 42/34 (76) | 61.8 ± 12.55 (76) | Jinan | Asia | Bios_Cancer (26) Bios_NAT (22) Salv_OSCC (28) | 0.6918 |
| Schmidt | 2014 | Swab (67) | 34/33 (67) | 53.33 ± 18.22 (67) | New York | North America | Swab_Cancer (15) Swab_Control (16) Swab_Health (19) Swab_OPMD (17) | 0.9139 |
| Zhang | 2019 | Swab (72) | NA (72) | NA (72) | Shanghai | Asia | Swab_Cancer (38) Swab_Control (34) | 0.6374 |
| zhao | 2017 | Swab (80) | NA (80) | NA (80) | Shanghai | Asia | Swab_Cancer (40) Swab_Control (40) | 1 |
| Anna | 2018 | Salv (17) | 8/2 (10); NA (7) | 66.1 ± 12.27 (10); NA (7) | Auckland | Oceania | Salv_OSCC (10) Salv_Health (7) | 0.4669 |
| Chen | 2021 | Salv (73) | Male (73) | 50.99 ± 11.29 (73) | Tainan | Asia | Salv_OSCC (27) Salv_Health (27) Salv_OPMD (19) | 0.4161 |
| Frank | 2022 | Salv (97) | 39/56 (95); NA (2) | 53.93 ± 15.82 (95); NA (2) | NA | NA | Salv_OSCC (19) Salv_Health (78) | <0.0001 |
| Lee | 2017 | Salv (369) | 333/36 (369) | 52.04 ± 12.45 (369) | Taichung | Asia | Salv_OSCC (123) Salv_Health (124) Salv_OPMD (122) | 0.9919 |
| MY Chen | 2021 | Salv (73 ) | Male (73) | 51.63 ± 14.25 (73) | Tainan | Asia | Salv_OSCC (45) Salv_OPMD (28) | 0.04662 |
| Torralba | 2021 | Salv (12) | NA (12) | NA (12) | Poland | Europe | Salv_OSCC (12) | 1 |
| DRP008029 | NA | Rins (44) | NA (44) | NA (44) | Australia | Oceania | Rins_OSCC (22) Rins_Health (22) | 1 |
| Sawant | 2021 | Rins (27) | NA (27) | 46.81 ± 14.11 (26); NA (1) | Mumbai | Asia | Rins_OSCC (9) Rins_Health (18) | 0.08326 |

[a]NA represents missing information in the corresponding category.
[b]N represents the total sample size in the data set.
[c]n represents the number of included samples in the corresponding category.
[d]Continuous variables were expressed as means ± standard deviations.

we maintained six categories (Table S1). Utilizing the Bray–Curtis distance at the genus level, we evaluated microbial diversity and applied permutational multivariate analysis of variance (PERMANOVA) to assess the impact of these six categories on the overall oral microbial composition. The findings revealed that the "Sample Type" category had the highest pseudo-F value and a comparatively high $R^2$ value, while the "Phenotype" category displayed the third-highest pseudo-F value and the second-highest $R^2$ value. Additionally, the "Geography" category exhibited the highest $R^2$ value. The influence of "Continent," "Sex," and "Age" categories on microbial composition was minimal (Fig. 1D). Due to high correlation between the "Geography" and "Study Name" categories (Cramer's V = 0.8592), the effect of the "Geography" category might be attributed to the substantial variances in microbial composition among different studies ($P < 0.001$, PERMANOVA $R^2 = 0.265$). Consequently, we included "Geography" as a random effect in the general linear model (Fig. 4E and F, Fig. 5B ). Considering the clinical significance and integrating the aforementioned findings, we primarily concentrated on the "Sample Type" and "Phenotype" categories during our analysis.

## Each sample type exhibits a distinct oral microbial community

Regarding the "Sample Type" category, current evidence indicates that particular sample types correspond to one or several unique niches in the oral cavity, with each niche supporting a complex and distinctive microbial community (18, 19, 31). This suggests that different sample types may harbor distinct microbial communities. To validate this, we investigated the oral microbial features in diverse sample types concerning microbial composition, diversity, and network. The overall microbial composition, grouped by "Sample Type" category, was calculated at both phylum and genus levels. At the phylum level, *Bacillota*, *Pseudomonadota*, *Bacteroidota*, *Actinomycetota*, and *Fusobacteriota* dominate the entire oral microbial composition (Fig. S1A). At the genus level, the oral microbial composition mainly consists of *Streptococcus*, *Veillonella*, *Prevotella*, *Fusobacterium*, *Porphyromonas*, *Actinomyces*, *Rothia*, *Haemophilus*, *Granulicatella*, and *Anaerococcus*, which comprise the top ten genera (Fig. 2A; Data S2).

We also assessed the microbial diversity of different sample types. Regarding alpha diversity, there were substantial differences among nearly all paired sample types (Fig. S1C). This states a substantial variation in the richness, abundance, and evenness of the microbiota across different sample types. Principal coordinate analysis (PCoA) based on Bray–Curtis distance at the genus level (Fig. 2B; Fig. S1B) indicated that the microbiome can be classified into five clusters based on the "Sample Type" category ($P < 0.001$). The Bray–Curtis dissimilarity indices further corroborated this difference, demonstrating that the beta diversity of Bios and Swab samples is higher than that of Salv and Rins samples, excluding Plaq samples (Fig. 2C).

Network relationships among microbes can assess the microbial community structure. Co-occurrence networks for the four sample types (excluding Plaq samples due to insufficient sample size) were separately constructed at the genus level (Fig. 2D). As anticipated, these networks display a power-law distribution, revealing scale-free characteristics (Fig. S1D). Network topology attribute analysis showed that the networks of Bios and Swab samples have lower negative correlations and higher modularity relative to Salv and Rins samples (Table S2). We also examined whether differences exist in the theoretical distribution patterns of node features in the four networks and confirmed this using the Kolmogorov–Smirnov (KS) test (Fig. 2E ; S1E). Furthermore, we conducted a robustness test on the co-occurrence networks of the four sample types using natural connectivity as an indicator, revealing variations in the stability of the microbial community structure across sample types (Fig. S1F). In summary, there are huge differences in the microbial community structure among these four sample types.

Finally, we discovered that microbes at the phylum to genus levels exhibit widespread differences across the five sample types (Fig. 2F). Specifically, *Pseudomonadales*, *Fusobacteriaceae*, and *Fusobacterium* are enriched in Bios samples; *Pasteurellales*, *Porphyromonadaceae*, and *Porphyromonas* are enriched in Swab samples; *Bacillota*,

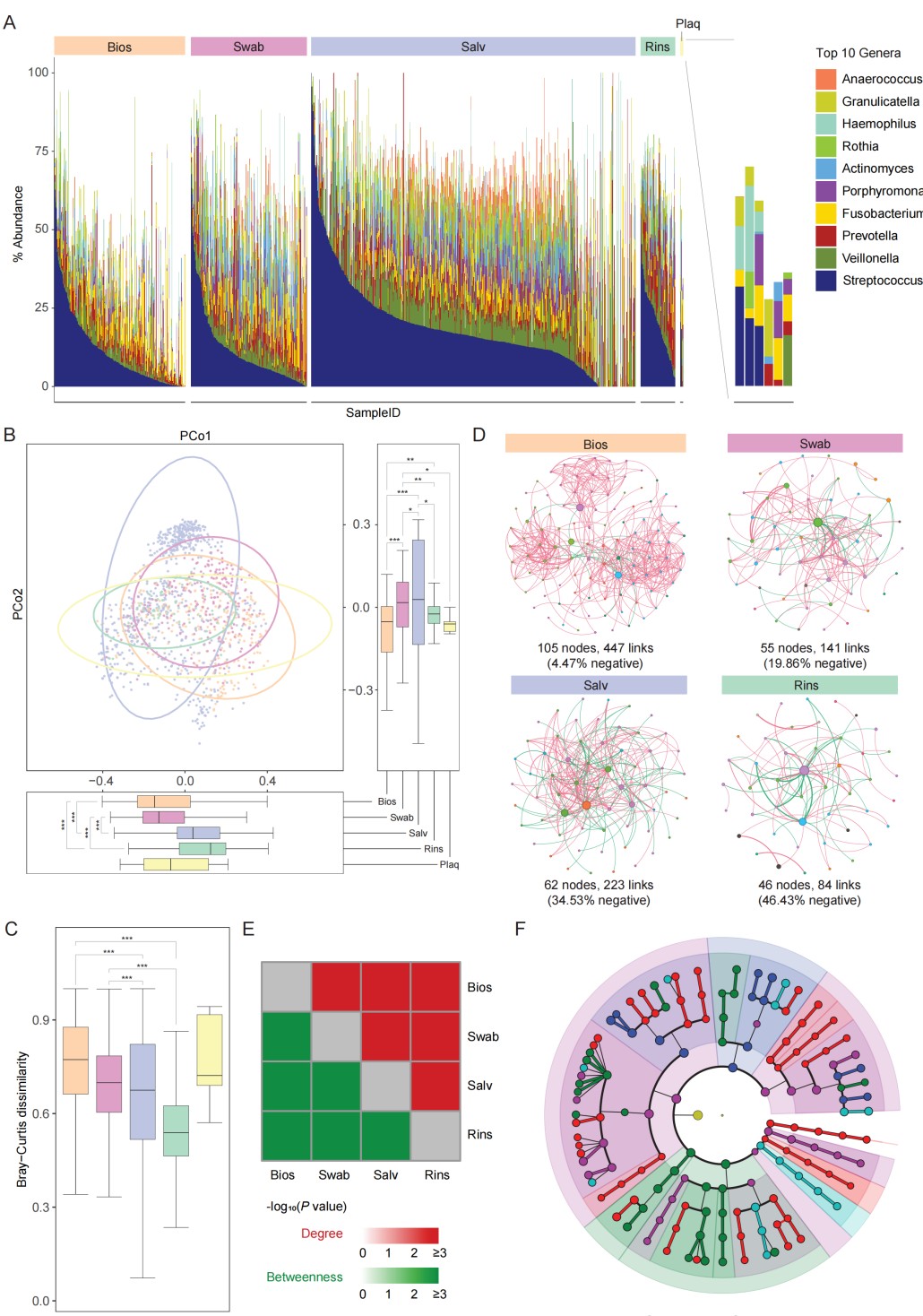

**FIG 2** Comparative analysis of oral microbial composition, diversity, and network across sample types (A) Composition bar plot depicting the top 10 bacterial genera in the overall sample across five sample types. (B) PCoA revealing significant microbiome differences among the five sample types. (C) Assessment of beta diversity across different sample types using Bray–Curtis dissimilarity indices extracted from the Bray–Curtis distance matrix. (D) Co-occurrence network constructed using the SparCC algorithm. Positive correlations are denoted in red, whereas negative correlations are displayed in green. Edge thickness positively correlated with the correlation between individual nodes. Nodes within the same module share the same color. The size of each node corresponds to its betweenness centrality. (E) Bootstrap resampling (10,000 replicates) estimates the theoretical distribution of node features (degree and betweenness) in the co-occurrence network. The KS test is employed

**Fig 2 (Continued)**

to compare their distribution between pairwise sample types. (F) The cladogram represents the characteristic microbes of the five sample types. Significance was determined using the linear discriminant analysis (LDA) effect size (LEfSe) method, with an LDA score >3 and $P < 0.05$. Unless otherwise specified, data are stated as boxplots, which show median, 25th, and 75th percentiles and $1.5 \times$ IQR. Pairwise comparisons were performed using the Wilcoxon rank-sum test (* $P < 0.05$, ** $P < 0.01$, and *** $P < 0.001$). Abbreviations: PCoA, principal coordinate analysis; SparCC, The Sparse Correlations for Compositional data; KS, Kolmogorov–Smirnov.

*Veillonellaceae*, and *Veillonella* are enriched in Salv samples; *Bacteroidia*, *Prevotellaceae*, and *Prevotella* are enriched in Rins samples; while *Pseudomonadota*, *Fusobacteriales*, and *Fusobacteria* are enriched in Plaq samples.

Taken together, these results indicate that each sample type possesses a distinct microbial community, emphasizing the importance of not mixing or confusing different sample types during analysis.

## Enhanced microbiome differences in Bios and Swab samples across phenotypes

Upon confirming the presence of distinct oral microbiome in each sample type, we focused on examining the microbiome differences across diverse phenotypes within the specified sample type to determine the most appropriate sample type for investigating OSCC progression. Utilizing the Bray–Curtis dissimilarity indices at the genus level, we detected significant distinctions in beta diversity among various phenotypes for all sample types. Additionally, the beta diversity exhibited an escalating trend in tandem with OSCC progression (Fig. 3A). Furthermore, we appraised alpha diversity, revealing conspicuous disparities in Evenness, Observed_OTUs, and Shannon index in Bios and Swab samples across multiple phenotypes, differing from the outcomes in Salv and Rins samples (Fig. 3B). We also performed PERMANOVA analysis on four sample types (excluding Plaq samples for the reason mentioned in the preceding section) and determined that the impact of the "Phenotype" category on microbial composition was more pronounced in Bios and Swab samples compared to Salv and Rins samples (Fig. 3C).

Taken together, the findings suggest that the microbiome of diverse phenotypes in Bios and Swab samples exhibit greater variability compared to those in Salv and Rins samples. This implies an enhanced potential for elucidating the progression of OSCC using Bios and Swab samples as the basis for analysis.

## Microbiome in Cancer and NAT groups exhibits similarity, while distinct from that of the FEP group

In Bios samples, each phenotype displays distinct microbiomes. To better understand the dynamic changes in the oral microbiome during OSCC progression, we thoroughly examined the relationships among the microbiome of different phenotypes. The microbial composition across these phenotypes was first meticulously characterized. At the phylum level, the composition of Bios samples is fairly consistent with that of the overall samples (Fig. S2A). At the genus level, Bios samples' microbial composition primarily comprises *Streptococcus*, *Fusobacterium*, *Prevotella*, and other genera (Fig. 4A). The Bray–Curtis distance at the genus level was utilized to assess the similarity of microbiomes among the three phenotypes in Bios samples. This revealed a closer proximity between Cancer and NAT groups and a greater distance from the FEP group (Fig. 4B through D ). Moreover, we separately constructed co-occurrence networks for the three phenotypes and calculated nodes' within-module (Zi) and among-module (Pi) connectivities. The co-occurrence network of the Cancer group exhibits the most complex structure but possesses the lowest negative correlation and the fewest key nodes within its network (Fig. S2B and C, Table S3).

Previous research has suggested that OSCC progression often coincides with alterations in specific microbes and microbial functions (32). Consequently, we focused

on the differential genera and KEGG pathways among phenotypes. We assigned Geography as a random effect and constructed a general linear model on MaAsLin2 to discern significantly different genera and pathways between pairwise comparisons of the three phenotypes (Data S3 and S4). The findings indicate that, in comparison to the FEP group, the Cancer and NAT groups exhibit significant enrichment of OSCC-associated genera and depletion of oral health-related genera (21, 32, 33). Specifically, we initially identified a comparison between NAT and FEP groups where the NAT group exhibited enrichment of eight genera, including *Dialister*, *Gemella*, and *Staphylococcus*, while the FEP group had enrichment only in *Streptococcus*. Second, in the comparison between Cancer and FEP groups, the Cancer group displayed an abundance of *Prevotella*, *Capnocytophaga*, and *Bulleidia*, with a depletion of *Megamonas* and *Rothia*. Notably, the abundance change trends of these identified genera are relatively consistent between Cancer and NAT groups. Functionally, the Cancer group exhibits enrichment of cancer and immunity-related pathways, such as prostate cancer, amoebiasis, and antigen processing and presentation pathways, compared to the FEP group. In contrast, the apoptosis pathway, which inhibits OSCC onset and progression, was found to be depleted in the NAT group when compared to the FEP group. Unexpectedly, the Cancer group did not show any enrichment of genera compared to the NAT group, but rather

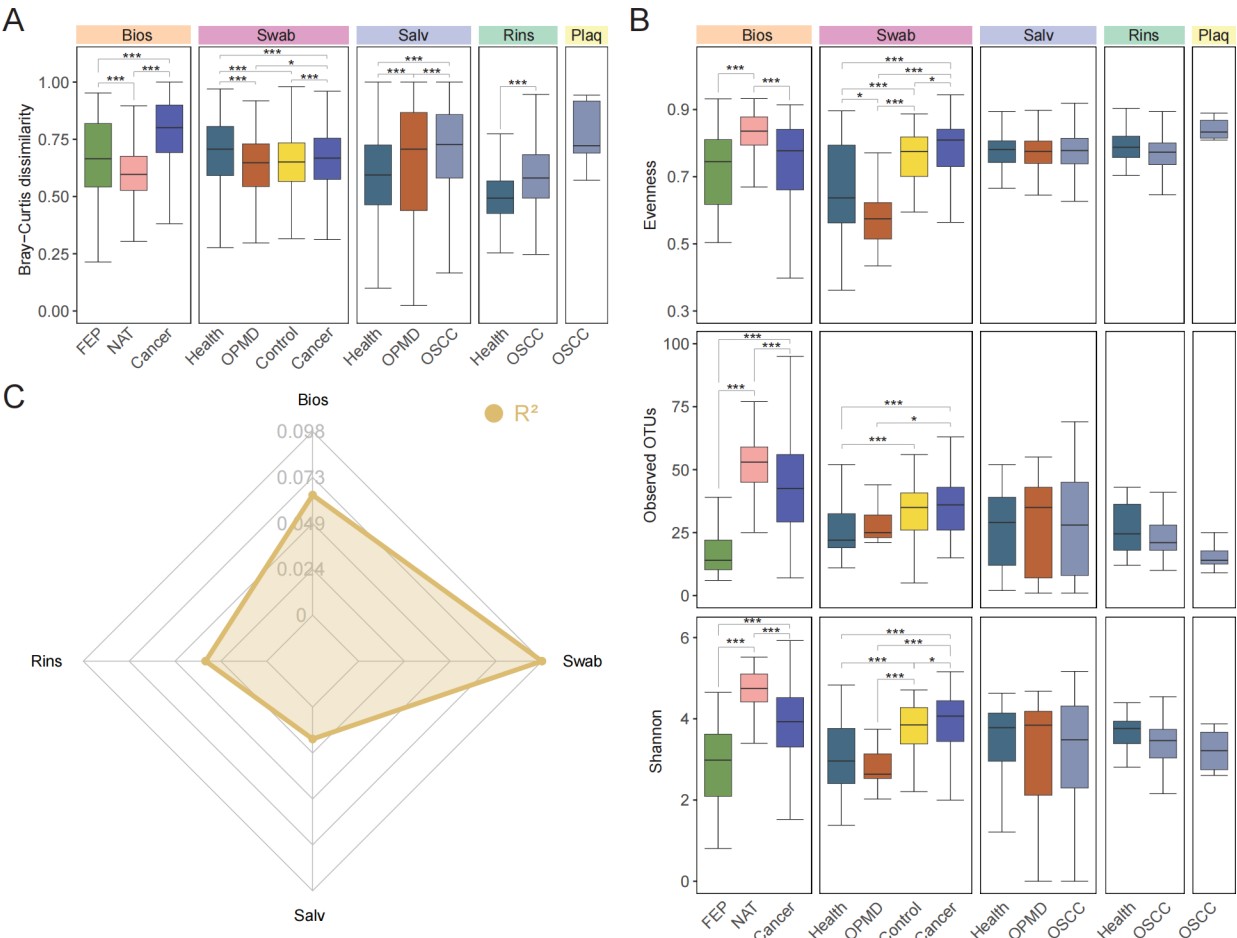

**FIG 3** Comparisons of alpha and beta diversity across phenotypes in various sample types (A) Distribution of Bray–Curtis dissimilarity indices for various phenotypes within each sample type. (B) Alpha diversity of the microbiome in different phenotypes within each sample type evaluated using Evenness, Observed_OTUs, and Shannon index. (C) Radar plot illustrating the impact of the "Phenotype" category on the variability of oral microbiome across sample types ($P < 0.05$, PERMANOVA, Bray–Curtis distance). Unless otherwise specified, data are stated as boxplots, which show median, 25th, and 75th percentiles and 1.5 × IQR. Pairwise comparisons were performed using the Wilcoxon rank-sum test (* $P < 0.05$, ** $P < 0.01$, *** $P < 0.001$). Abbreviations: PERMANOVA, permutational multivariate analysis of variance.

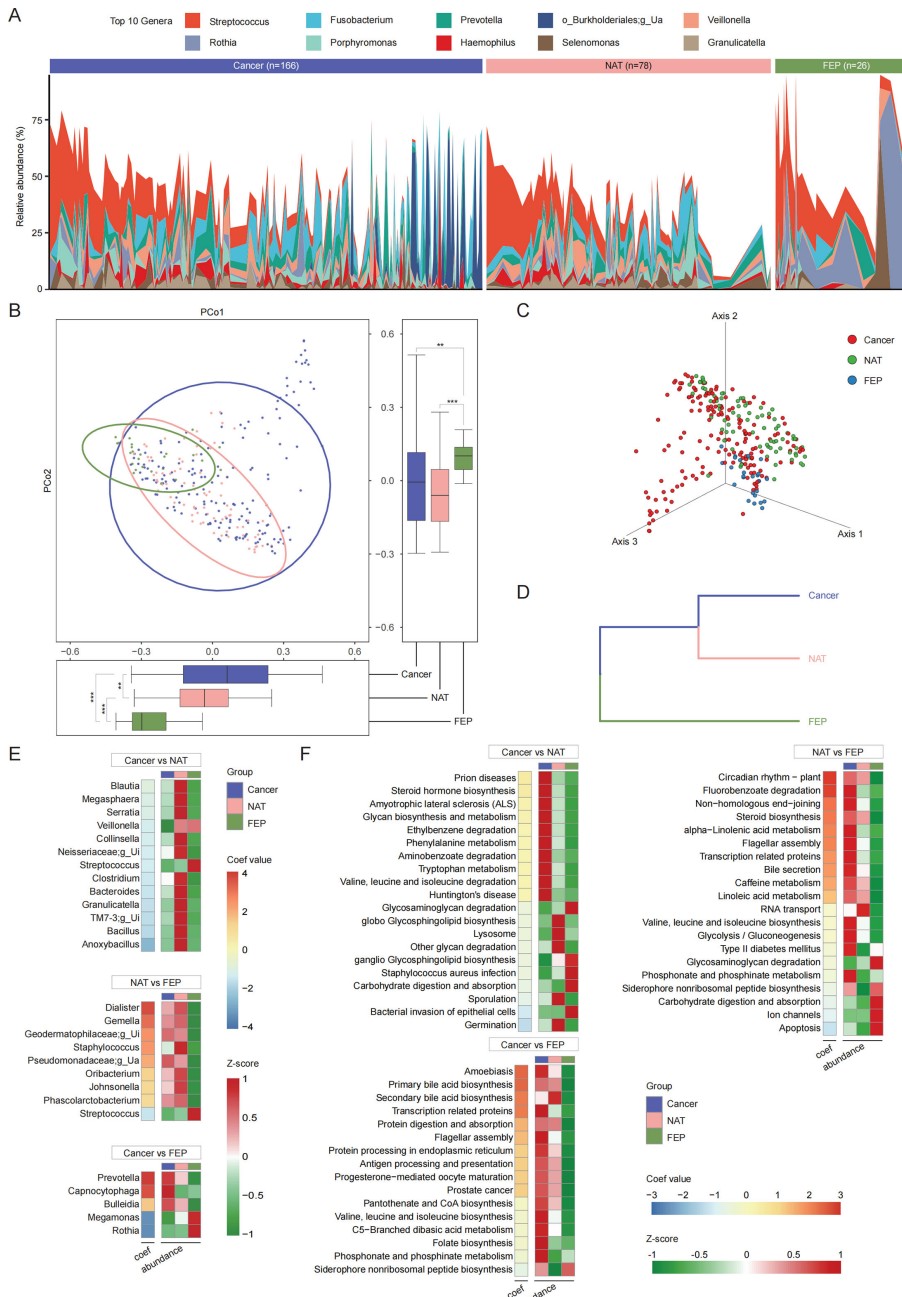

**FIG 4** Identification of relationships between microbiomes across different phenotypes (A) Stacked bar plot displaying the relative abundance of the top 10 bacterial genera within each phenotype. (B) 2D-PCoA ($P < 0.001$, PERMANOVA) of the three phenotypes in Bios samples. (C) 3D-PCoA ($P < 0.001$, PERMANOVA) in Bios samples. (D) Clustering the three phenotypes of Bios samples utilizing the unweighted pair-group method with arithmetic mean (UPGMA). (E) Representation of correlation coefficients (left) and relative abundance (right) of significantly different bacterial genera between pairwise comparisons of the three phenotypes in Bios samples. Significantly different genera were determined using MaAsLin2 adjusting for Geography, with an adjusted $P < 0.05$ as the cut-off for significance. The heatmap shows a positive correlation with the values of the correlation coefficients. Concurrently, the Z-score standardizes the abundance of various genera within the three phenotypes. (F) Illustration of correlation coefficients (left) and relative abundance (right) of significantly different KEGG pathways between pairwise comparisons of the three phenotypes in Bios samples (depicting only the TOP10 pathways for each phenotype within each comparison pair). Other settings remain consistent with those in (E). Unless otherwise specified, data are stated as boxplots, which show median, 25th, and 75th percentiles and 1.5 × IQR. Pairwise comparisons were performed using the Wilcoxon rank-sum test (* $P < 0.05$, ** $P < 0.01$, and *** $P < 0.001$). Abbreviations: PERMANOVA, permutational multivariate analysis of variance; g_Ui, g_Unidentified; g_Ua, g_Unassigned; MaAsLin2, The Microbiome Multivariable Associations with Linear Models; KEGG, Kyoto Encyclopedia of Genes and Genomes.

displayed depletion of 13 genera, most of which are health-related. Correspondingly, at the pathway level, we identified enrichment of the *Staphylococcus aureus* infection pathway and bacterial invasion of epithelial cell pathway in the NAT group (Fig. 4E and F).

Taken together, our investigation of Bios samples disclosed the similarity between microbiome in Cancer and NAT groups and their divergence from that of the FEP group. Furthermore, the results of significantly different genera and KEGG pathways provide additional support for the observed outcomes.

## Construction of a cross-platform diagnostic model based on random forest

Swab samples display two distinct characteristics: first, considerable disparities exist among the microbiomes of different phenotypes (Fig. 3A through C); second, the sampling method is straightforward and facile. Collectively, these attributes facilitate the construction of diagnostic models. Greater heterogeneity among microbiomes enhances the performance of diagnostic models used for classification (34). Consequently, an in-depth examination of the similarities between the microbiomes of the four pheno-types in swab samples was conducted. The assessment revealed that the microbiomes of Cancer and Control groups are substantially distanced from those of Health and OPMD groups, corroborating our observations from Bios samples (Fig. 5A). In light of these findings and with clinical requirements, a diagnostic model was developed to discern between Cancer and Health groups.

Utilizing the expanded Swab data set (Cancer group: $n = 105$; Health group: $n = 184$), we employed MaAsLin2 to construct a general linear model with Geography as a random effect, identifying 20 genera with significant differences (Data S5). Subsequently, based on these 20 genera, we implemented six classical machine learning algorithms—Gaussian naive Bayes (GNB), support vector machine (SVM), logistic regression (Log-Reg), k-nearest neighbors (KNN), random forest (RF), and XGBoost (XGB)—to develop preliminary diagnostic models. The tenfold cross-validation was used to assess the classification performance of the models. The XGB model exhibited the best perform-ance with AUC as the evaluation criterion, followed by RF, while LogReg displayed the lowest performance (Fig. 5B; Table S4). Taking into account other classification metrics of accuracy, F1-score, precision, and recall, we ultimately selected the RF model for subsequent predictions. We employed MeanDecreaseAccuracy to gauge the importance of the 20 genera and, based on the tenfold cross-validation of the random forest, ascertained five genera (*Catonella*, *Prevotella*, *Capnocytophaga*, *Peptostreptococcus*, and *Streptococcus*) as the optimal feature set for the diagnostic model (Fig. 5C). Utilizing these five microbial genera, we reconstructed the RF model and evaluated its performance using receiver operating characteristic (ROC) analysis, achieving an AUC of 0.918 for the test set (Fig. 5D). Upon applying this model to an external data set sequenced via the NextSeq platform, the AUC reached 0.849, indicating robust generalization performance (Fig. 5E).

Taken together, we constructed an effective diagnostic model based on the RF algorithm that distinguishes between Cancer and Health groups. The model comprises five genera and is applicable to another sequencing platform.

## DISCUSSION

We conducted the meta-analysis by extensively collecting independent research and integrating their oral microbiome data, aiming to gain a more accurate understanding of universal microbiome characteristics during OSCC progression. Currently, there is a lack of systematic comparisons involving the oral microbiome from various sample types (16, 17). Our results revealed that the microbiome of different sample types have unique characteristics and warrant separate analyses. Furthermore, it remains unclear which sample type is optimal for investigating the oral microbiome during OSCC progression. We observed notable differences in the microbiome across different phenotypes in Bios and Swab samples, whereas the microbiome in Salv (35), Rins, and Plaq samples demonstrated relative consistency among distinct phenotypes. We speculate that the

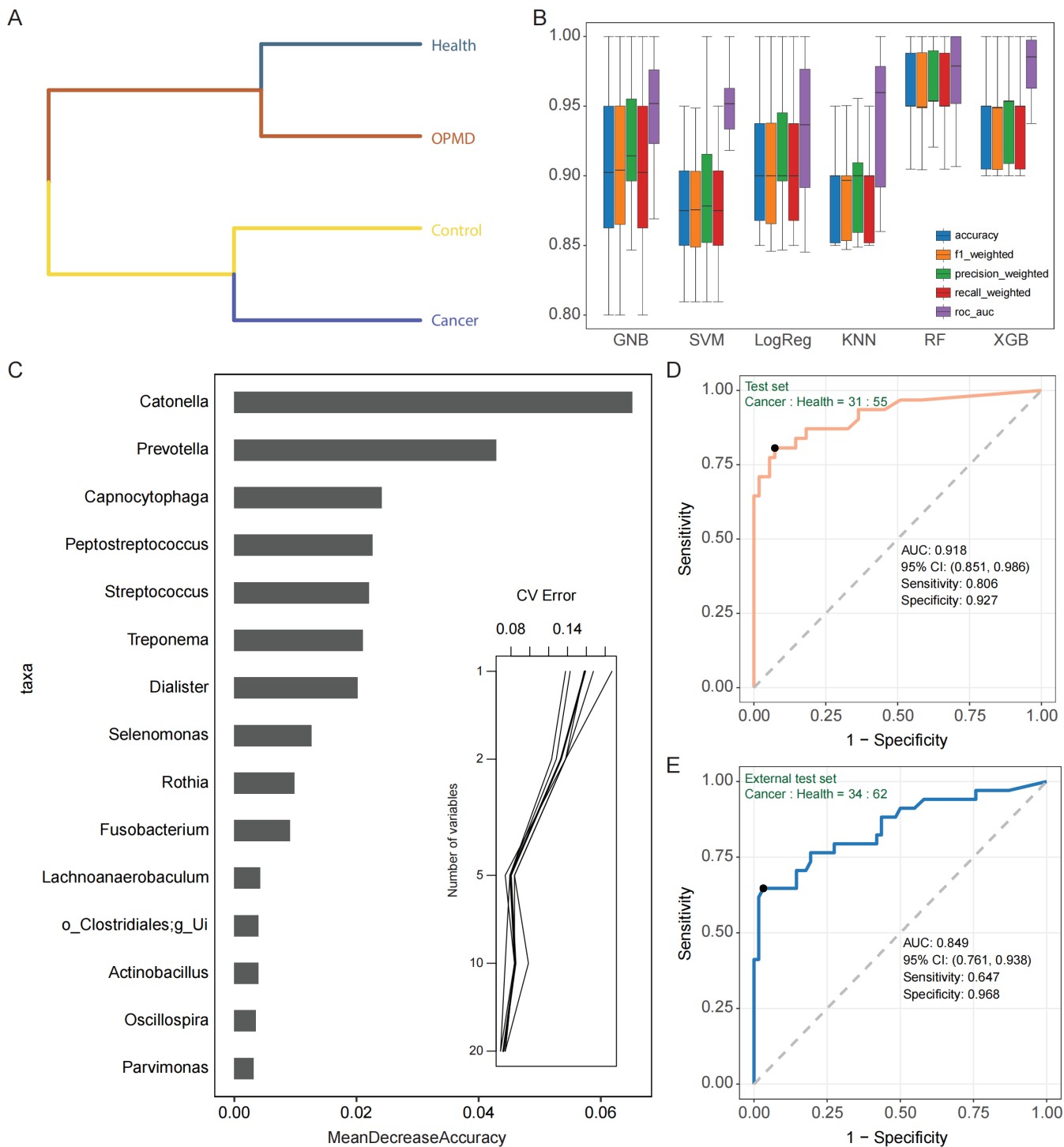

**FIG 5** Distinguishing Cancer and Health groups based on bacterial genera (A) UPGMA clustering tree (Bray–Curtis distance) for the four phenotypes in Swab samples. (B) Evaluation of the performance of six machine learning algorithms in distinguishing Cancer and Health groups utilizing five classification metrics. (C) The top 20 genera are ordered in descending importance concerning their contribution to the accuracy of the RF model (left). Subsequently, five microbial biomarkers were determined as the optimal markers for the RF model via tenfold cross-validation (right). (D) ROC analysis for the RF model distinguishing the Cancer group from the Health group in the test set, with a sample size ratio of 7:3 between the training set and test set. (E) ROC analysis of the established RF model discriminating between the Cancer group and the Health group within the external test set (NextSeq; Cancer group: $n = 34$; Health group: $n = 64$). Unless otherwise specified, data are stated as boxplots, which show median, 25th, and 75th percentiles and $1.5 \times$ IQR. Pairwise comparisons were performed using the Wilcoxon rank-sum test (* $P < 0.05$, ** $P < 0.01$, and *** $P < 0.001$). Abbreviations: UPGMA, unweighted pair-group method with arithmetic mean; g_Ui, g_Unidentified; RF, random forest; ROC, receiver operating characteristic.

direct exposure of the oral cavity to the external environment may lead to a more significant influence of environmental factors on Salv, Rins, and Plaq samples (18, 34, 36). Thus, Bios and Swab samples might be more suitable for studying the oral microbiome in OSCC.

Extensive exploration has been conducted concerning the associations between the microbiome of various phenotypes. In Bios samples, the microbiomes of the Cancer and NAT groups are more similar to each other, while both differ from the FEP group's microbiome. Correspondingly, in Swab samples, the microbiomes of the Cancer and Control groups, as well as the OPMD and Health groups, exhibit a closer relationship (Fig. 4D and Fig. 5A). Given that the Bios samples for the Cancer and NAT groups and the Swab samples for the Cancer and Control groups are all obtained from OSCC patients, a plausible explanation is that this reflects microbiome communication between adjacent niches in OSCC patients' oral cavities (18, 37–39). However, as this explanation is based on preliminary studies, further comparative investigations between OSCC patients and healthy individuals are needed to confirm this observation. Consequently, in contrast to genome studies focused on the host, further investigation is required concerning the customary approach of utilizing peri-tumor tissues and their surface mucosae as controls in microbiome research emphasizing host–environment interactions (40, 41). Moreover, from a more specific perspective, we examined the relationships between the three phenotypes in Bios samples, concentrating on significantly different genera and KEGG pathways (Fig. 4E and F). This provides novel insights into the dynamic changes of the OSCC oral microbiome. First, the Cancer and NAT groups may harbor a similar "cancer microbiome." Second, compared to the Cancer group, the NAT group has not entirely lost the healthy genera it carries, and its microbial function may still primarily involve early invasion and infection. Lastly, the FEP group has the potential to serve as a control for exploring OSCC progression. Considering that some studies have aimed to target the microbiome in OSCC treatment, it is essential to enhance therapeutic efficacy while minimizing adverse effects (42). Key genera such as *Porphyromonas*, and particularly invasive *Treponema* and *Campylobacter*—all well-known initiators of inflammation—were upregulated in the Cancer and NAT groups (with *Porphyromonas* upregulated in NAT compared to Cancer) (12, 33). However, these findings did not reach statistical significance and were thus not highlighted in the Fig. 4E (Source Data). One possible explanation is that this may support the "driver-passenger" model, whereby tumor initiation is triggered by "driver" bacteria and subsequently succeeded by "passenger" bacteria in the tumor micro-environment (43).

Our study has some limitations. First, during the literature inclusion phase, data for certain specific phenotypes were available from only one primary study, which might weaken the generalizability of our conclusions. Second, when processing raw data, we mapped sequencing reads to the full-length 16S rRNA gene sequence to ensure comparability of sequencing data amplified from different gene regions. Despite multiple studies validating the functionality of this method, it inevitably introduces a certain level of false positives (44). In constructing diagnostic models, one of our goals was to achieve a balanced sample size between the Cancer and Health groups. Moreover, we sought to supplement the Health group by including all accessible independent studies, thereby minimizing subjective selection bias. However, this resulted in a slight imbalance in the sample size between the Cancer and Health groups (Cancer group: $n = 105$; Health group: $n = 184$). Additionally, during the external evaluation phase of the model, only one additional new data set was generated in the publicly available repositories, with an imbalance in the sample sizes for the Cancer and Health groups (Cancer group: $n = 34$; Health group: $n = 62$). Since these imbalanced proportions were both less than twofold, constituting moderate imbalance, we did not make further adjustments (45, 46). Furthermore, while meta-analysis, as a large-scale standardized approach, allows for comprehensive data integration, it inherently reduces the ability to account for the biological specificities of individuals, potentially missing unique characteristics in rare cases (47). Lastly, Future taxonomic revisions, such as the ongoing

reclassification of the *Prevotella* genus, may affect model performance (48). However, our results indicate that the current framework exhibits adaptability to such adjustments (Fig. S3gg2_External test set). As taxonomic systems continue to evolve, dynamic updates to taxonomic annotations and feature selection will be essential to ensure the model's sustained reliability.

Recently, the unique role of intracellular bacteria within tumor tissue has generated widespread interest in the presence and function of tumor-associated bacteria (49, 50). Due to the direct exposure of the oral cavity to the external environment, the interactions between the oral microbiome and their surroundings may extend far beyond our previous understanding. In addition, the oral cavity is characterized by convenient sampling and abundant bacterial colonization. Consequently, the most significant contribution of this study is the investigation of the research system concerning the direct interaction between microbiota and tumors, specifically in OSCC. This provides comprehensive and summarized information on the cancer microbiome for various sample types during cancer progression. Furthermore, recent advances in sequencing technologies (51), such as 2bRAD-M, have significantly enhanced resolution at the species taxonomy level, offering new perspectives for understanding the role of the microbiome in OSCC (52, 53).

Although this study primarily focuses on the tumor microbiome characterization and OSCC diagnosis, the effects of therapeutic treatments on the oral microbiome require further investigation. For instance, standard treatments like surgery and radiotherapy, along with platinum-based chemotherapy for cancer cell radiosensitization, can substantially alter the oral microbiome (33, 54). Studies have shown that radiotherapy increases the abundance of cariogenic bacteria such as *Streptococcus mutans* and *Lactobacillus*, as well as *Staphylococcus*, *Enterococcus*, and the opportunistic pathogen *Candida albicans*, while reducing commensal bacteria like *Neisseria* (55). These microbial shifts are further complicated by treatment side effects like xerostomia, which creates conditions favorable for *Lactobacillus* proliferation that persist for months after radiotherapy (33). Therefore, additional multicenter studies with diverse patient populations are emergently needed to better understand the interactions between OSCC treatments and the oral microbiome.

## MATERIALS AND METHODS

### Incorporation of public data

The Supplementary Material offers a detailed account of the entire process of incorporating public data (Fig. S3; Data S6). The inclusion of data sets is divided into two stages. The first stage, known as the primary analysis (Stage 1), was conducted until mid-March 2022. We searched PubMed using the keywords "(('oral squamous cell carcinoma') OR ('OSCC')) AND ((('16S') OR ('metagenome')) OR ('microbiome'))" and simultaneously searched the SRA and ENA databases using the keyword "oral squamous cell carcinoma." Without limiting hypervariable regions, sample types, or OSCC phenotypes, we consolidated the search results from these three databases and contacted the corresponding and first authors of data sets with incomplete information to obtain the original sequencing data or metadata. The data sets preliminarily included in the meta-analysis met the following key criteria: (i) 16S rRNA gene sequencing was performed using the Illumina MiSeq platform (ii); relatively complete metadata were available.

The second stage (Stage 2) entailed constructing a diagnostic model. By October 2022, we incorporated two studies' 16S rRNA gene data sets to supplement the Health group in swab samples. Concurrently, we collected a newly published swab data set containing both Cancer and Health groups from the period between the first and second stages, which served as an external independent test set. During these two stages, most of the metadata information of the included studies was manually added. In each metadata category, missing information was annotated as "NA," and information not included was annotated as "NU." Abbreviations in the metadata can be found in the Data

S7. In accordance with the ICD-11 standard, codes 2B60-2B66 are classified as OSCC. For age classification, patients under 45 years of age were defined as "younger," those aged 45–60 years as "middle," and those over 60 years old as "older." In tumor staging, some sample information annotations adopted the AJCC 7th edition standard. Taking the time span into account, we conducted an evaluation to align with the AJCC 8th edition standard (samples that could not be matched were discarded). The final meta-analysis data set exhibits the relationships among the three metadata categories of "Sample Type," "Phenotype," and "Sample Name" (Table S5).

## 16S data preprocessing

In this study, a standardized pipeline in QIIME 2 (version 2020.11.0) was employed to process all 16S rRNA gene data sets (56). The demultiplexed sequencing reads of each data set were denoised using the DADA2 protocol (57). Subsequently, the VSEARCH allpairs_global algorithm was employed to map all reads to full-length 16S rRNA sequences (58). A custom Naive Bayes classifier was trained on the Greengenes 13_8 99% operational taxonomic units (OTUs) to assign taxonomy to each data set (59). This pipeline minimized limitations associated with sequencing platforms and hypervariable regions, enabling direct comparisons between studies. Samples containing fewer than 1,000 sequences after quality filtering and OTU assignment were excluded from the analysis.

## Biodiversity analysis

The QIIME 2 diversity plugin was employed to calculate three alpha diversity metrics, including Evenness, Observed_OTUs, and Shannon index, as well as the Bray–Curtis distance matrix. The PCoA visualization was accomplished using the Bray–Curtis distance matrix, and the Bray–Curtis dissimilarity indices were extracted. PERMANOVA analysis (999 permutations) was performed using the vegan package (Version 2.6–4) in R.

## Microbial correlation network analysis

The Sparse Correlations for Compositional data (SparCC) algorithm was employed to construct correlation networks with the SpiecEasi package (Version 1.1.2) in R (60). After calculating pseudo $P$ matrices based on 1,000 bootstrap iterations, only edges with an absolute correlation value >0.3 and a $P < 0.05$ were retained, and all isolated nodes were removed. The undirected networks were visualized, and the related network topological attributes were calculated in Gephi (version 0.10.1) software. Multiple node topological features were extracted from the constructed SparCC network, and the theoretical distribution of node features was estimated via 10,000 bootstrap resampling. On the nodes of the co-occurrence network, betweenness centrality and node degree were calculated and ranked by importance. Subsequently, nodes were sequentially removed from the network, simulating genera loss based on the importance ranking. The natural connectivity changes following node removal were assessed to evaluate the network's connectivity capabilities after sustaining a certain degree of damage, thus indirectly reflecting the stability of the microbial community structure.

## Identification of significantly different bacterial genera

The Microbiome Multivariable Associations with Linear Models (MaAsLin2) package (Version 1.10.0) was utilized in R to determine differentially abundant bacteria at the genus level between distinct phenotypes by employing unscaled relative abundance (61). The Q value package was incorporated in MaAsLin2 for multiple testing correction (BH-FDR correction, $q$-value) of 0.25. We set the significance criteria as $q$-value <0.05, mean relative abundance >0.001, and mean prevalence >0.2. In line with the developers' recommendations, other parameters remained unadjusted.

## Identification of significantly different pathways

For microbiome functional predictions, the Phylogenetic Investigation of Communities by Reconstruction of Unobserved States (PICRUSt) method was utilized (62). We identified the differential pathways between phenotypes using MaAsLin2, incorporating geography as a random effect, in line with the section above. In the presentation of the results, only the third level of Kyoto Encyclopedia of Genes and Genomes (KEGG) pathways was retained (63).

## Construction and evaluation of diagnostic models

The data set was randomly divided into training and testing sets with a 7:3 ratio. Six widely applied machine learning models were used from Python's Scikit-learn package (Version 1.2.1), namely, GNB, SVM, LogReg, KNN, RF, and XGB, for model construction (64). During training, a tenfold cross-validation method was adopted, and multiple metrics were calculated to gauge the overall performance of each model. The Caret package (Version 6.0–94) in R was then employed to independently build and optimize the final RF model (65). Ultimately, the constructed models were assessed on the test set and external independent test set using ROC curves, and AUC values were calculated.

## Statistical analysis

Pairwise comparison was performed using the two-sided Wilcoxon rank-sum test (Mann–Whitney U test). The Kolmogorov–Smirnov test was utilized to compare the distribution of node features in co-occurrence networks across four distinct sample types. Additionally, the Benjamini–Hochberg false discovery rate correction method was applied to adjust $P$ for multiple testing. All statistical analyses in this study were conducted in R (Version 4.2.2).

## ACKNOWLEDGMENTS

This work was supported by the Sichuan Science and Technology Program of China [grant number 2023NSFSC1460] and the National Natural Science Foundation of China [grant number 32100927].

Z.W. conceived the study, performed bioinformatics analysis, drafted and revised the manuscript; Y.C. supported bioinformatics analysis, completed the majority of visualization work in the study, and revised the manuscript; H.L. provided critical feedback on the research and revised the manuscript; Y.Y. provided critical feedback on the research and supervised the research; H.Y. conceived the study, supported bioinformatics analysis, supervised the research, and revised the manuscript.

## AUTHOR AFFILIATIONS

[1]West China Biomedical Big Data Center, West China Hospital, Sichuan University, Chengdu, China
[2]Med-X Center for Informatics, Sichuan University, Chengdu, China
[3]Department of Stomatology, Shanghai Sixth People's Hospital Affiliated to Shanghai Jiao Tong University School of Medicine, Shanghai, China
[4]State Key Laboratory of Oral Diseases & National Clinical Research Center for Oral Diseases, West China Hospital of Stomatology, Sichuan University, Chengdu, China
[5]Department of Prosthodontics, West China Hospital of Stomatology, Sichuan University, Chengdu, China

## AUTHOR ORCIDs

Zizheng Wang  http://orcid.org/0000-0002-6967-0630
Yuan Yue  http://orcid.org/0000-0002-3076-4971

## DATA AVAILABILITY

In this study, all utilized sequencing data were obtained from publicly available repositories (Data S6), and the raw data used for subsequent analysis were preserved within the Source Data. Both data sets, along with supplementary materials such as tables and figures, are accessible at Figshare (https://doi.org/10.6084/m9.figshare.28227341.v1). Additionally, the constructed diagnostic model can be found at https://github.com/eirainal/diagnostic-model.git.

## ADDITIONAL FILES

The following material is available online.

### Open Peer Review

**PEER REVIEW HISTORY (review-history.pdf).** An accounting of the reviewer comments and feedback.

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
