## [Reviewer comments · Microbiology Spectrum]

Microbiology Spectrum

Exploring oral microbiome in oral squamous cell carcinoma across environment-associated sample types

Zizheng Wang, Yilong Chen, Haoning Li, Yuan Yue, and Haopeng Yu

Corresponding Author(s): Haopeng Yu, Sichuan University West China Hospital

Review Timeline:

Submission Date:	April 3, 2024
Editorial Decision:	September 20, 2024
Revision Received:	December 11, 2024
Editorial Decision:	December 19, 2024
Revision Received:	January 2, 2025
Accepted:	January 7, 2025

Editor: Zhenjiang Xu

Reviewer(s): Disclosure of reviewer identity is with reference to reviewer comments included in decision letter(s). The following individuals involved in review of your submission have agreed to reveal their identity: Georg Conrads (Reviewer #2); Pierre Le Bars (Reviewer #3)

Transaction Report:

DOI: <https://doi.org/10.1128/spectrum.00852-24>

Re: Spectrum00852-24 (Exploring oral microbiome in oral squamous cell carcinoma across environment-associated sample types)

Dear Prof. Haopeng Yu:

Thank you for the privilege of reviewing your work. Below you will find my comments, instructions from the Spectrum editorial office, and the reviewer comments.

Revision Guidelines

Sincerely,
Zhenjiang Xu
Editor
Microbiology Spectrum

Reviewer #2 (Comments for the Author):

In their meta-analysis entitled „Exploring oral microbiome in oral squamous cell carcinoma (OSCC) across environment-associated sample types" Wang et al. provide a comprehensive overview of the specimen-specific oral microbiome during various OSCC progression stages, potentially enhancing early detection and treatment."

General comments:

Introduction: nice, but after line 73 one or two sentences are missing about the etiology of OSCC. Otherwise, OSCC could be mis-interpreted as infectious disease. Example: It has been reported that the majority of cases can be related to tobacco use and/or heavy alcohol consumption (Ref). Other possible risk factors include viral infections (Ref), poor oral hygiene (Ref), and infection with *Candida* species. Some forms of leukoplakic lesions have long been recognized as an independent risk factor for carcinoma.

Lines 74ff: "These hypotheses can be summarized as dysregulation in the oral microbiome may contribute to the development of OSCC via mechanisms ..." Grammar; please re-write.

Materials and Methods: no comments; please re-check all details.

Results

Line 168ff: "At the phylum level, Firmicutes, Proteobacteria, Bacteroidetes, Actinobacteria, and Fusobacteria...", please mind the new taxonomy: Bacillota (Firmicutes), Pseudomonadota (Proteobacteria), Bacteroidota (Bacteroidetes), Actinomycetota (Actinobacteria), and Fusobacteriota (Fusobacteria).

Line 254: "with a depletion of Megamonas": species of Megamonas are not regarded as oral species (see eHOMD database and publications). Could this be a mis-identification? Check/discuss.

Line 295ff: "[we] ascertained five genera (Catonella, Prevotella, Capnocytophaga, Peptostreptococcus, Streptococcus) as the optimal feature set for the diagnostic model." This sounds very promising. However, please keep in mind how many species of very different type and pathogenicity are within every genus. A mechanistic explanation is hard to draw from these NGS findings. Reservation: and what if the genera are spit into new genera (like currently with Prevotella). You say your model is "robust"; I say it is actually very vulnerable if taxonomy is changing.

Discussion:

In your discussion, I am missing a section about the impact of changes in taxonomy on robustness of your model.

Otherwise a nice meta-analysis.

This reviewer has a basic bioinformatics knowledge but cannot judge about most of the methods use. However, they are looking familiar/appropriate and the results plausible. Mind the new nomenclature and the risk that some of your taxon identifications (as in the case of Megamonas") might be mis- or over-interpretation. On the other hand, where on the list of key-genera are Porphyromonas, and especially invasive Treponema / Campylobacter; all three well known initiators of inflammation? Please discuss.

Reviewer #3 (Comments for the Author):

Manuscript # Spectrum00852-24

Current Revision # 0

Submission Date 2024-04-03 09:52:07

Title

Exploring oral microbiome in oral squamous cell carcinoma across environment-associated sample types

Manuscript Type Research Article

Special

Section N/A

Corresponding Author

Prof. Haopeng Yu (Sichuan University West China Hospital)

Contributing Authors

Dr. Zizheng Wang , Dr. Yilong Chen , Haoning Li , Dr. Yuan Yue , Prof. Haopeng Yu (corr-auth)

Review .

OSCC remains complex regarding both its etiology and its growth. Locally inflammation and microbial imbalance are mentioned. Historically, three hypotheses confront each other regarding the microbiota. Is it the pre-existing microbiota that can modify the microenvironment and initiate the process of carcinogenesis? Or is it the oncological pathology itself that causes the appearance of pathogens? Or both at the same time. The contribution of this article through a meta-analysis is an attempt to answer these questions. However, the lack of homogeneity of the populations studied, protocols concerning samples, and microbial analyzes constitute obstacles and harm the reliability of the results. Certain bacteria such as *F. nucleatum* and *P. gingivalis* as well as *Streptococcus* species are present upstream and downstream of the OSCC. Also, several studies have focused on the existence of a pre-existing microbiota in healthy patients that can predispose to OSCC, without yet providing an answer. The purpose of these investigations, based on a hypothesis, is to determine the link between OSCC and the oral microbiota. The goal is to develop less invasive and more cost-effective ways to detect cancer.

However, the complexity of interactions of the oral microbiome (bacteria, viruses, fungi, etc.) with host cells still remains opaque.

Current research concerns the genomes and metabolites of microorganisms present in the environment, rather than the identification of different bacterial species.

Finally, the article proposes, from the swab, a diagnosis based on a RF algorithm that distinguishes the Cancer and Health groups. The model, however, is limited to five genera and does not capture variabilities at the species level. The definitive diagnosis can only be made from a biopsy.

Several publications in 2024 on the subject should be included in the bibliography in order to update the article.

Line 64: abbreviated (ROC) and (UAC) must be translated in full.

Line 77 : More specific OSCC articles are preferable such as:

Wang J, Gao B. Mechanisms and Potential Clinical Implications of Oral Microbiome in Oral Squamous Cell Carcinoma. *Curr Oncol.* 2023 Dec 28;31(1):168-182. doi: 10.3390/curroncol31010011. PMID: 38248096; PMCID: PMC10814288.

Line 89: However, large-scale standardized approaches have the disadvantage of minimizing the biological specificities of each individual.

Line 164: more recent references suggest new perspectives.such..

(Lim 2024) A new microbiological investigation of plaque using Type IIB Restriction-site Associated DNA for Microbiome (2bRAD-M) sequencing. provides access to the species taxonomy level for biofilms.

Independent way, providing quantitative and qualitative data by demonstrating and resolving species-level taxonomy .Sun Z, Huang S, Zhu P, et al. Species-resolved sequencing of low-biomass or degraded microbiomes using 2bRAD-M. *Genome Biol.* 2022;23(1):1-22. doi: 10.1186/s13059-021-02576-9

Line 326: Reference 12 concerns healthy patients, extrapolation to OSCC patients remains hypothetical.

Line 329: Among other things, a recent reference is missing which seems to partly respond to the environmental conditions in the presence of an OSCC .Such... Cai L, Zhu H, Mou Q, Wong PY, Lan L, Ng CWK, Lei P, Cheung MK, Wang D, Wong EWY, Lau EHL, Yeung ZWC, Lai R, Meehan K, Fung S, Chan KCA, Lui VWY, Cheng ASL, Yu J, Chan PKS, Chan JYK, Chen Z. Integrative analysis reveals associations between oral microbiota dysbiosis and host genetic and epigenetic aberrations in oral cavity squamous cell carcinoma. *NPJ Biofilms Microbiomes.* 2024 Apr 8;10(1):39. doi: 10.1038/s41522-024-00511-x. PMID: 38589501; PMCID: PMC11001959.

Line 342: Here too it is an extrapolation between the hypothesis put forward by a publication on colorectal cancer and OSCC.

In conclusion, the objective of this research is missing. Indeed, the microbiota is considered by some authors as a target of OSCC treatment, participating in traditional treatment. The objective is to improve therapeutic effects while reducing adverse effects.

Recent bibliography.

Salivary biomarkers for early detection of oral squamous cell carcinoma (OSCC) and head/neck squamous cell carcinoma (HNSCC): A systematic review and network meta-analysis

Shahnawaz Khijmatgar, Josh Yong, Nicole Rüksamen, Felice Lorusso, Pooja Rai, Niccolo Cenzato, Franscesca Gaffuri, Massimo Del Fabbro, Gianluca Martino Tartaglia

Jpn Dent Sci Rev. 2024 Dec; 60: 32-39. Published online 2023 Dec 21. doi: 10.1016/j.jdsr.2023.10.003

PMCID: PMC107

Metabolomics and metagenomics reveal the impact of $\gamma\delta$ T inhibition on gut microbiota and metabolism in periodontitis-promoting OSCC Wei Wei, Jing Li, Boyu Tang, Ye Deng, Yan Li, Qianming Chen *mSystems.* 2024 Feb; 9(2): e00777-23.

Published online 2024 Jan 23. doi: 10.1128/msystems.00777-23 PMCID:

Porphyromonas gingivalis suppresses oral squamous cell carcinoma progression by inhibiting MUC1 expression and remodeling the tumor microenvironment

Zhou Lan, Ke-Long Zou, Hao Cui, Yu-Yue Zhao, Guang-Tao Yu

Mol Oncol. 2024 May; 18(5): 1174-1188. Published online 2023 Sep 13. doi: 10.1002/1878-0261.13517

PMCID: PMC11076995

Cai L, Zhu H, Mou Q, Wong PY, Lan L, Ng CWK, Lei P, Cheung MK, Wang D, Wong EWY, Lau EHL, Yeung ZWC, Lai R,

Meehan K, Fung S, Chan KCA, Lui VWY, Cheng ASL, Yu J, Chan PKS, Chan JYK, Chen Z. Integrative analysis reveals associations between oral microbiota dysbiosis and host genetic and epigenetic aberrations in oral cavity squamous cell carcinoma. NPJ Biofilms Microbiomes. 2024 Apr 8;10(1):39. doi: 10.1038/s41522-024-00511-x. PMID: 38589501; PMCID: PMC11001959.

Heller G, Fuereder T, Grandits AM, Wieser R. New perspectives on biology, disease progression, and therapy response of head and neck cancer gained from single cell RNA sequencing and spatial transcriptomics. Oncol Res. 2023 Nov 15;32(1):1-17. doi: 10.32604/or.2023.044774. PMID: 38188682; PMCID: PMC10767240.

Reviewer #4 (Comments for the Author):

Wang et al. present a meta-analysis with 1,255 samples from OSCC-related 16S rRNA gene amplicon sequencing datasets. They demonstrate correlations between microbiome composition, sample types, and cancer stages. The manuscript is generally well-written, but some concerns need to be addressed.

Major

- Though all 16S rRNA gene sequencing datasets were performed with the Illumina MiSeq platform, how about the amplicon regions and the primers? Were they the same? Different regions of the 16S rRNA might have variable discriminative power. L406 mentioned: "This approach overcame limitations associated with sequencing platforms and hypervariable regions". It's not clear what the "This" points to. Besides, how was the batch effect handled?

- L444, how's the dataset been randomly divided? I mean it would be better to replicate the dividing-training-testing procedure multiple times (with different random seeds) and see how the overall and average performance would be, not just once. I know it would take more effort.

Minor

- All citations are behind periods, e.g., L71.
- L119, Extra comma: "Habitat,".
- L404, is the source code of the custom Naive Bayes classifier available?
- L418, typo: "iteratiox`ns"
- L462, are there more data which are only available upon request? Will the author give after publication? It's quite common these years that the authors do not reply upon request, just as the authors asked for data from previous researchers.

Wang et al. present a meta-analysis with 1,255 samples from OSCC-related 16S rRNA gene amplicon sequencing datasets. They demonstrate correlations between microbiome composition, sample types, and cancer stages. The manuscript is generally well-written, but some concerns need to be addressed.

Major

- Though all 16S rRNA gene sequencing datasets were performed with the Illumina MiSeq platform, how about the amplicon regions and the primers? Were they the same? Different regions of the 16S rRNA might have variable discriminative power. L406 mentioned: "This approach overcame limitations associated with sequencing platforms and hypervariable regions". It's not clear what the "This" points to. Besides, how was the batch effect handled?

- L444, how's the dataset been randomly divided? I mean it would be better to replicate the dividing-training-testing procedure multiple times (with different random seeds) and see how the overall and average performance would be, not just once. I know it would take more effort.

Minor

- All citations are behind periods, e.g., L71.

- L119, Extra comma: "Habitat,".

- L404, is the source code of the custom Naive Bayes classifier available?

- L418, typo: "iteratiox`ns"

- L462, are there more data which are only available upon request? Will the author give after publication? It's quite common these years that the authors do not reply upon request, just as the authors asked for data from previous researchers.

Zhenjiang Xu
Editor,
Microbiology Spectrum.

Dear editor and reviewers,

We wish to resubmit our revised manuscript for publication in *Microbiology Spectrum*, under the title, “Exploring oral microbiome in oral squamous cell carcinoma across environment-associated sample types”. The reference number for our initial submission is Spectrum00852-24R1.

Thank you for your letter and for the reviewers’ comments concerning our manuscript entitled “Exploring oral microbiome in oral squamous cell carcinoma across environment-associated sample types”. The comments were very helpful for revising and improving our manuscript. We have read the comments carefully and have made corrections in the revised manuscript that we hope will meet with approval. Revised text is marked in red. Our responses to the reviewers’ comments can be found below. **Additionally, in accordance with the journal’s formatting requirements, we have added an 'Importance' section to the manuscript, found on line 68-77.**

We hope that our manuscript is now acceptable for publication. If not, then we look forward to working with you and the reviewers to move this manuscript closer to publication in *Microbiology Spectrum*.

Yours Sincerely,
Haopeng Yu, Ph.D.,
Med-X Center for Informatics, Sichuan University,
West China Biomedical Big Data Center, West China Hospital/West China School of Medicine,
Chengdu 610041, China
Email: yuhaopeng@wchscu.cn

Yuan Yue, D.D.S, Ph.D.,
The State Key Laboratory of Oral Diseases & National Clinical Research Center for Oral Diseases,
Department of Prosthodontics, West China Hospital of Stomatology,
NO.14, 3rd Section of Ren Min Nan Rd. 610041, Sichuan, China
Email: hxkqyueyuan@163.com

Responses to the reviewers' comments:

Reviewer #2 (Comments for the Author):

In their meta-analysis entitled „Exploring oral microbiome in oral squamous cell carcinoma (OSCC) across environment-associated sample types" Wang et al. provide a comprehensive overview of the specimen-specific oral microbiome during various OSCC progression stages, potentially enhancing early detection and treatment."

General comments:

#1 Introduction: nice, but after line 73 one or two sentences are missing about the etiology of OSCC. Otherwise, OSCC could be mis-interpreted as infectious disease. Example: It has been reported that the majority of cases can be related to tobacco use and/or heavy alcohol consumption (Ref). Other possible risk factors include viral infections (Ref), poor oral hygiene (Ref), and infection with *Candida* species. Some forms of leukoplakic lesions have long been recognized as an independent risk factor for carcinoma.

Our response:

Thank you for the reviewer's guidance. We have added content regarding the etiology of OSCC in the Introduction section to avoid potential misinterpretation by readers:

“Tobacco use and/or heavy alcohol consumption have been shown to increase the risk of OSCC (4-7). Other possible risk factors include viral infections, poor oral hygiene, and *Candida* species infection (8). Additionally, certain forms of leukoplakia have long been recognized as independent risk factors for carcinoma development (9). Dysbiosis of the oral microbiome has also been implicated in OSCC progression (10).” (line 83-87)

#2 Lines 74ff: "These hypotheses can be summarized as dysregulation in the oral microbiome may contribute to the development of OSCC via mechanisms ..." Grammar; please re-write.

Our response:

This sentence has been revised to:

“These hypotheses suggest that dysregulation in the oral microbiome may contribute to the development of OSCC through mechanisms involving inflammatory responses, cellular invasion, immune modulation, and toxic metabolite production (14, 15).” (line 88-91)

Materials and Methods:

#3 no comments; please re-check all details.

Our response:

We have re-checked all details in this section.

Results:

#4 Line 168ff: "At the phylum level, Firmicutes, Proteobacteria, Bacteroidetes, Actinobacteria, and Fusobacteria...", please mind the new taxonomy: Bacillota (Firmicutes), Pseudomonadota

(Proteobacteria), Bacteroidota (Bacteroidetes), Actinomycetota (Actinobacteria), and Fusobacteriota (Fusobacteria).

Our response:

Thank you for your correction! The new taxonomy has been applied. (line 182-183; line 213-215)

#5 Line 254: "with a depletion of Megamonas": species of Megamonas are not regarded as oral species (see eHOMD database and publications). Could this be a mis-identification? Check/discuss.

Our response:

We carefully examined the original dataset and found that the result, "Megamonas depletion in the Cancer group compared to the FEP group," was primarily contributed by Sarkar et al.'s study (Source Data; Fig. 4B-D). In their paper [1], Sarkar et al. stated:

"At the genus level, 22 taxa including Serratia, Anoxybacillus, Stenotrophomonas, Sutterella, Actinomyces, Bacillus, Lysobacter, Paenibacillus, Ammoniphilus, Bifidobacterium, Megamonas, Collinsella, Brevibacillus, Megasphaera, Blautia, Methylobacterium, Prevotella_9, Roseburia, Phenylbacterium, Pseudopropionibacterium, Parabacteroides, and Anaerobacillus were significantly declined in the OSCC lesions as compared to the healthy controls (Figure 4C)."

The consistency between their findings and our results suggests that our study faithfully reflects the original data.

While *Megamonas* is predominantly recognized as a gut microbiome genus, its detection in oral studies such as Sarkar et al.'s work raises interesting possibilities. Although databases like eHOMD have not categorized *Megamonas* as a typical oral genus, this does not definitively exclude its presence. Emerging evidence supports the detection of *Megamonas* in oral niches under specific conditions [2-3].

Further research is needed to confirm the presence and role of *Megamonas* in the oral microbiome and its potential implications in OSCC.

Reference:

[1] Sarkar P, Malik S, Laha S, Das S, Bunk S, Ray JG, Chatterjee R, Saha A. Dysbiosis of Oral Microbiota During Oral Squamous Cell Carcinoma Development. *Front Oncol.* 2021 Feb 23;11:614448. doi: 10.3389/fonc.2021.614448IF: 3.5 Q2 . PMID: 33708627IF: 3.5 Q2 ; PMCID: PMC7940518

[2] Woodall CA, Hammond A, Cleary D, Preston A, Muir P, Pascoe B, Sheppard SK, Hay AD. Oral and gut microbial biomarkers of susceptibility to respiratory tract infection in adults: A feasibility study. *Heliyon.* 2023 Jul 28;9(8):e18610. doi: 10.1016/j.heliyon.2023.e18610. PMID: 37593638; PMCID: PMC10432180.

[3] Xu S, Xiang C, Wu J, Teng Y, Wu Z, Wang R, Lu B, Zhan Z, Wu H, Zhang J. Tongue Coating Bacteria as a Potential Stable Biomarker for Gastric Cancer Independent of Lifestyle. *Dig Dis Sci.* 2021 Sep;66(9):2964-2980. doi: 10.1007/s10620-020-06637-0. Epub 2020 Oct 12. PMID: 33044677.

#6 Line 295ff: "[we] ascertained five genera (Catonella, Prevotella, Capnocytophaga, Peptostreptococcus, Streptococcus) as the optimal feature set for the diagnostic model." This sounds very promising. However, please keep in mind how many species of very different type and pathogenicity are within every genus. A mechanistic explanation is hard to draw from these NGS findings. Reservation: and what if the genera are spit into new genera (like currently with Prevotella). You say your model is "robust"; I say it is actually very vulnerable if taxonomy is changing.

Our response:

We appreciate the reviewer's concern regarding the impact of taxonomic changes on the robustness of our diagnostic model. To address this, we reannotated the original dataset using the Greengene2 database (gg2_External test set.csv), an updated taxonomic system that incorporates recent taxonomic adjustments. Using the same dataset and analytical methods, we retrained and validated the diagnostic model with the newly annotated features. The results demonstrated consistent diagnostic performance under both Greengene1 and Greengene2 annotations, with only slight variations (e.g., AUC: 0.849 for Greengene1, Fig. 5E, vs. 0.836 for Greengene2, Fig. S3gg2_External test set). Sensitivity and specificity showed minor trade-offs, indicating the model's robustness across different taxonomic frameworks.

We acknowledge that taxonomic systems will continue to evolve, as evidenced by the ongoing reclassification within the genus *Prevotella*. Our findings suggest that the model can adapt to such changes, likely due to the ensemble nature of the random forest algorithm, which integrates predictions across multiple decision trees. This feature allows the model to maintain accuracy by leveraging stable features despite certain taxonomic shifts. As taxonomic systems continue to evolve, dynamic updates to taxonomic annotations and feature selection will be essential to ensure the model's sustained reliability.

Fig. S3gg2_External test set

Discussion:

#7 In your discussion, I am missing a section about the impact of changes in taxonomy on robustness of your model. Otherwise a nice meta-analysis.

Our response:

In response to the reviewer's suggestion, we have expanded the discussion on the limitations of our study, including the potential impact of future taxonomic changes on the robustness of the model:

“Lastly, Future taxonomic revisions, such as the ongoing reclassification of the *Prevotella* genus, may affect model performance (48). However, our results indicate that the current framework exhibits adaptability to such adjustments (Fig. S3gg_External test set). As taxonomic systems continue to evolve, dynamic updates to taxonomic annotations and feature selection will be essential to ensure the model's sustained reliability.” (line 380-384)

#8 (1) This reviewer has a basic bioinformatics knowledge but cannot judge about most of the methods use. However, they are looking familiar/appropriate and the results plausible. Mind the new nomenclature and the risk that some of your taxon identifications (as in the case of Megamonas") might be mis- or over-interpretation. (2) On the other hand, where on the list of key-genera are Porphyromonas, and especially invasive Treponema / Campylobacter; all three well known initiators of inflammation? Please discuss.

Our response:

(1) The reviewer's questions have been addressed in our responses to Reviewer #5, #6 and #7.

(2) Thank you for the reviewer's concern. We have adjusted the relevant writing to fully address the raised issue and have added the corresponding data in the Source Data.

“ Key genera such as *Porphyromonas*, and particularly invasive *Treponema* and *Campylobacter*—all well-known initiators of inflammation—were upregulated in the Cancer and NAT groups (with *Porphyromonas* upregulated in NAT compared to Cancer) (12, 33). However, these findings did not reach statistical significance and were thus not highlighted in the Fig. 4E (Source Data). ” (line 355-359)

Reviewer #3 (Comments for the Author):

Review:

OSCC remains complex regarding both its etiology and its growth. Locally inflammation and microbial imbalance are mentioned.

Historically, three hypotheses confront each other regarding the microbiota. Is it the pre-existing microbiota that can modify the microenvironment and initiate the process of carcinogenesis? Or is it the oncological pathology itself that causes the appearance of pathogens? Or both at the same time. The contribution of this article through a meta-analysis is an attempt to answer these questions. However, the lack of homogeneity of the populations studied, protocols concerning samples, and microbial analyzes constitute obstacles and harm the reliability of the results. Certain bacteria such as *F. nucleatum* and *P. gingivalis* as well as *Streptococcus* species are present upstream and downstream of the OSCC. Also, several studies have focused on the existence of a pre-existing microbiota in healthy patients that can predispose to OSCC, without yet providing an answer. The purpose of these investigations, based on a hypothesis, is to determine the link between OSCC and the oral microbiota. The goal is to develop less invasive and more

cost-effective ways to detect cancer.

However, the complexity of interactions of the oral microbiome (bacteria, viruses, fungi, etc.) with host cells still remains opaque. Current research concerns the genomes and metabolites of microorganisms present in the environment, rather than the identification of different bacterial species.

Finally, the article proposes, from the swab, a diagnosis based on a RF algorithm that distinguishes the Cancer and Health groups. The model, however, is limited to five genera and does not capture variabilities at the species level. The definitive diagnosis can only be made from a biopsy.

Details:

#9 Line 64: abbreviated (ROC) and (UAC) must be translated in full.

Our response:

Thank you for the reviewer's suggestion. The issue of incorrect abbreviations has been addressed, and the content has been corrected as follows:

“We developed a diagnostic model using Random Forest algorithm on Swab samples, achieving an area under the receiver operating characteristic curve (AUC) of 0.918.” (line 62-64)

#10 Line 77 : More specific OSCC articles are preferable such as:

Wang J, Gao B. Mechanisms and Potential Clinical Implications of Oral Microbiome in Oral Squamous Cell Carcinoma. *Curr Oncol.* 2023 Dec 28;31(1):168-182. doi: 10.3390/currenco131010011. PMID: 38248096; PMCID: PMC10814288.

Our response:

More specific OSCC references have been cited. (line 84)

#11 Line 89: However, large-scale standardized approaches have the disadvantage of minimizing the biological specificities of each individual.

Our response:

Thank you for the reviewer's concern. We have added relevant content to the limitations section of the Discussion:

“Furthermore, while meta-analysis, as a large-scale standardized approach, allows for comprehensive data integration, it inherently reduces the ability to account for the biological specificities of individuals, potentially missing unique characteristics in rare cases (47).” (line 377-380)

#12 Line 164: more recent references suggest new perspectives.such..

(Lim 2024) A new microbiological investigation of plaque using Type IIB Restriction-site Associated DNA for Microbiome (2bRAD-M) sequencing. provides access to the species taxonomy level for biofilms. Independent way, providing quantitative and qualitative data by demonstrating and resolving species-level taxonomy .Sun Z, Huang S, Zhu P, et al. Species-resolved sequencing of low-biomass or degraded microbiomes using 2bRAD-M. *Genome*

Our response:

More specific references have been cited, and we have added the following content:

“Furthermore, recent advances in sequencing technologies (51), such as 2bRAD-M, have significantly enhanced resolution at the species taxonomy level, offering new perspectives for understanding the role of microbiome in OSCC (52, 53).” (line 393-396)

#13 Line 326: Reference 12 concerns healthy patients, extrapolation to OSCC patients remains hypothetical.

Our response:

Thank you for the reviewer’s correction. We have revised the writing and added additional references to address findings specifically in OSCC patients:

“a plausible explanation is that this reflects microbiome communication between adjacent niches in OSCC patients' oral cavities (18, 37-39). However, as this explanation is based on preliminary studies, further comparative investigations between OSCC patients and healthy individuals are needed to confirm this observation.” (line 340-343)

#14 Line 329: Among other things, a recent reference is missing which seems to partly respond to the environmental conditions in the presence of an OSCC .Such... Cai L, Zhu H, Mou Q, Wong PY, Lan L, Ng CWK, Lei P, Cheung MK, Wang D, Wong EWY, Lau EHL, Yeung ZWC, Lai R, Meehan K, Fung S, Chan KCA, Lui VWY, Cheng ASL, Yu J, Chan PKS, Chan JYK, Chen Z. Integrative analysis reveals associations between oral microbiota dysbiosis and host genetic and epigenetic aberrations in oral cavity squamous cell carcinoma. NPJ Biofilms Microbiomes. 2024 Apr 8;10(1):39. doi: 10.1038/s41522-024-00511-x. PMID: 38589501; PMCID: PMC11001959.

Our response:

The missing reference has been cited. (line 346)

#15 Line 342: Here too it is an extrapolation between the hypothesis put forward by a publication on colorectal cancer and OSCC.

Our response:

Thank you for the reviewer’s correction. We have revised the writing as follows:

“One possible explanation is that this may support the "driver-passenger" model, whereby tumor initiation is triggered by “driver” bacteria and subsequently succeeded by “passenger” bacteria in the tumor micro-environment (43).” (line 359-361)

#16 In conclusion, the objective of this research is missing. Indeed, the microbiota is considered by some authors as a target of OSCC treatment, participating in traditional treatment. The objective is to improve therapeutic effects while reducing adverse effects.

Our response:

Thank you for the reviewer’s insight. We have added the following content:

“Considering that some studies have aimed to target the microbiome in OSCC treatment, it is essential to enhance therapeutic efficacy while minimizing adverse effects (42). Key genera such as *Porphyromonas*, and particularly invasive *Treponema* and *Campylobacter*—all well-known initiators of inflammation—were upregulated in the Cancer and NAT groups (with *Porphyromonas* upregulated in NAT compared to Cancer) (12, 33). However, these findings did not reach statistical significance and were thus not highlighted in the Fig. 4E (Source Data).” (line 353-359)

Recent bibliography:

#17 Several publications in 2024 on the subject should be included in the bibliography in order to update the article.

1. Salivary biomarkers for early detection of oral squamous cell carcinoma (OSCC) and head/neck squamous cell carcinoma (HNSCC): A systematic review and network meta-analysis. Shahnawaz Khijmatgar, Josh Yong, Nicole Rübsamen, Felice Lorusso, Pooja Rai, Niccolo Cenzato, Francesca Gaffuri, Massimo Del Fabbro, Gianluca Martino Tartaglia. *Jpn Dent Sci Rev.* 2024 Dec; 60: 32-39. Published online 2023 Dec 21. doi: 10.1016/j.jdsr.2023.10.003 PMID: 38107107

2. Metabolomics and metagenomics reveal the impact of $\gamma\delta$ T inhibition on gut microbiota and metabolism in periodontitis-promoting OSCC. Wei Wei, Jing Li, Boyu Tang, Ye Deng, Yan Li, Qianming Chen. *mSystems.* 2024 Feb; 9(2): e00777-23. Published online 2024 Jan 23. doi: 10.1128/msystems.00777-23 PMID: 38107107

3. *Porphyromonas gingivalis* suppresses oral squamous cell carcinoma progression by inhibiting MUC1 expression and remodeling the tumor microenvironment. Zhou Lan, Ke-Long Zou, Hao Cui, Yu-Yue Zhao, Guang-Tao Yu. *Mol Oncol.* 2024 May; 18(5): 1174-1188. Published online 2023 Sep 13. doi: 10.1002/1878-0261.13517 PMID: 38107107

4. Cai L, Zhu H, Mou Q, Wong PY, Lan L, Ng CWK, Lei P, Cheung MK, Wang D, Wong EWY, Lau EHL, Yeung ZWC, Lai R, Meehan K, Fung S, Chan KCA, Lui VWY, Cheng ASL, Yu J, Chan PKS, Chan JYK, Chen Z. Integrative analysis reveals associations between oral microbiota dysbiosis and host genetic and epigenetic aberrations in oral cavity squamous cell carcinoma. *NPJ Biofilms Microbiomes.* 2024 Apr 8;10(1):39. doi: 10.1038/s41522-024-00511-x. PMID: 38589501; PMID: PMC11001959.

5. Heller G, Fuereder T, Grandits AM, Wieser R. New perspectives on biology, disease progression, and therapy response of head and neck cancer gained from single cell RNA sequencing and spatial transcriptomics. *Oncol Res.* 2023 Nov 15;32(1):1-17. doi: 10.32604/or.2023.044774. PMID: 38188682; PMID: PMC10767240.

Our response:

Publications mentioned above have been cited.

Reviewer #4 (Comments for the Author):

Wang et al. present a meta-analysis with 1,255 samples from OSCC-related 16S rRNA gene amplicon sequencing datasets. They demonstrate correlations between microbiome composition, sample types, and cancer stages. The manuscript is generally well-written, but some concerns need to be addressed.

Major:

#18 - Though all 16S rRNA gene sequencing datasets were performed with the Illumina MiSeq platform, how about the amplicon regions and the primers? Were they the same? Different regions of the 16S rRNA might have variable discriminative power. L406 mentioned: "This approach overcame limitations associated with sequencing platforms and hypervariable regions". It's not clear what the "This" points to. Besides, how was the batch effect handled?

Our response:

Thank you for your insightful questions.

In our methodology, as detailed in the Methods section 7.2, we mapped reads to full-length 16S rRNA sequences using the VSEARCH allpairs_global algorithm. After mapping, the originally captured specific region sequences were replaced with the full-length 16S rRNA sequences. This substitution step standardized the sequence data across samples, enabling direct comparisons despite initial differences in targeted regions. Once we completed this replacement, we reran the Qiime2 pipeline to process the updated sequences, ensuring consistent analysis across the dataset. This approach effectively mitigates issues related to the variations in the hypervariable regions and sequencing platform differences.

The "This approach" mentioned in Line 406 refers specifically to the strategy outlined above. We have revised this sentence in the original text to prevent any misunderstanding.

Regarding batch effect handling, we employed PERMANOVA, which allowed us to assess and account for potential biases across different datasets. We further addressed the batch effect by including "Geography" as a random effect in our general linear model.

The detailed explanation can be found in the manuscript:

“Utilizing the Bray-Curtis distance at the genus level, we evaluated microbial diversity and applied Permutational multivariate analysis of variance (PERMANOVA) to assess the impact of these six categories on the overall oral microbial composition. The findings revealed that the "Sample Type" category had the highest pseudo-F value and a comparatively high R2 value, while the "Phenotype" category displayed the third-highest pseudo-F value and the second-highest R2 value. Additionally, the "Geography" category exhibited the highest R2 value. The influence of "Continent," "Sex," and "Age" categories on microbial composition was minimal (Fig. 1D). Due to high correlation between the 'Geography' and 'Study Name' categories (Cramer's V = 0.8592), the effect of the 'Geography' category might be attributed to the substantial variances in microbial composition among different studies ($p < 0.001$, PERMANOVA R2 = 0.265). Consequently, we included "Geography" as a random effect in the general linear model (Figs. 4E, F, 5B).” (line

159-170)

#19 - L444, how's the dataset been randomly divided? I mean it would be better to replicate the dividing-training-testing procedure multiple times (with different random seeds) and see how the overall and average performance would be, not just once. I know it would take more effort.

Our response:

(1) Thank you for raising this concern. During the initial dataset split (training set: testing set = 7:3), we used a random seed (921, corresponding to Zizheng Wang's birthday). The code snippet is as follows:

```
import pandas as pd
data=pd.read_csv("dataset1.csv")
array = data.values

X = array[:,0:20]
y = array[:,20]
from sklearn.model_selection import train_test_split
X_train, X_test, y_train, y_test = train_test_split(X, y, test_size=0.3, random_state=921)
```

(2) Following your suggestion, we conducted an additional analysis by performing ten independent splits using random seeds ranging from 1 to 10. The performance of the models was evaluated across all splits. Overall, the RF and XGB models achieved comparable results in terms of ROC-AUC values, with slight variations in their respective performances. However, the RF model consistently outperformed other models in most other evaluation metrics. These findings align with the conclusions presented in our manuscript and do not affect the main results. The detailed performance metrics have been included in the supplementary file (random_result.xlsx).

Minor:

#20 - All citations are behind periods, e.g., L71.

Our response:

Thank you for pointing out this formatting issue. We have revised the manuscript to ensure that all citations are placed before the periods.

#21 - L119, Extra comma: "Habitat,".

Our response:

Thank you for identifying this issue. We have removed the extra comma. (line 133)

#22 - L404, is the source code of the custom Naive Bayes classifier available?

Our response:

The source code for the plugin can be accessed at the following repository:

<https://github.com/qiime2/q2-feature-classifier>.

For further details about the workflow, please refer to:

1. <https://docs.qiime2.org/2024.5/tutorials/feature-classifier/>.

2. <https://docs.qiime2.org/2024.5/plugins/available/feature-classifier/fit-classifier-naive-bayes/>.

Reference:

Bokulich, N.A., Kaehler, B.D., Rideout, J.R. et al. Optimizing taxonomic classification of marker-gene amplicon sequences with QIIME 2's q2-feature-classifier plugin. *Microbiome* 6, 90 (2018). <https://doi.org/10.1186/s40168-018-0470-z>.

#23 - L418, typo: "iteratiox`ns"

Our response:

Thank you for pointing out the typographical error. We have corrected "iteratiox`ns" to "iterations" in the revised manuscript. (line 447)

#24 - L462, are there more data which are only available upon request? Will the author give after publication? It's quite common these years that the authors do not reply upon request, just as the authors asked for data from previous researchers.

Our response:

Thank you for raising this concern. Upon review, we identified that the diagnostic model had not yet been made publicly available. To address this, we have now ensured open access to the diagnostic model, which can be found at the following repository:

<https://github.com/eirainal/diagnostic-model.git>.

Correspondingly, we have made the following adjustments in the manuscript:

“In this study, all utilized sequencing data were obtained from publicly available repositories (Supplementary Data 6), and the raw data used for subsequent analysis were preserved within the Source Data. Additionally, the constructed diagnostic model is accessible at <https://github.com/eirainal/diagnostic-model.git>.” (line 488-492)

Re: Spectrum00852-24R1 (Exploring oral microbiome in oral squamous cell carcinoma across environment-associated sample types)

Dear Prof. Haopeng Yu:

Thank you for the submission of your revision. Based on Reviewer #3's feedback, your revised manuscript still requires some changes and clarifications.

Revision Guidelines

Sincerely,
Zhenjiang Xu
Editor
Microbiology Spectrum

Reviewer #2 (Comments for the Author):

no further comments; well done

Reviewer #3 (Comments for the Author):

The authors have made corrections and provided clarifications on several points. However, the article does not sufficiently discuss the effects on the oral microbiome of the different treatments for OSCC. Many parameters, such as the host's diet, side effects of surgery and radiotherapy, and the administration of antibiotics can unbalance the microbiome. Similarly, local consequences such as the development of oral mucositis and dry mouth can also disrupt the microbial balance.

Most early or recurrent squamous cell carcinomas are treated with a surgical and radiotherapeutic approach, while, when implemented, chemotherapy (in the form of a platinum-based drug), allows radiosensitization of the cancer cells in the oral cavity.

Radiotherapy leads to an increase in the number of cariogenic bacteria such as *Streptococcus mutans* and *Lactobacillus*, as well as *Staphylococcus*, *Enterococcus* and the fungus *Candida albicans*, on the other hand there is a reduction in commensal microorganisms such as *Neisseria*. Species such as *Lactobacillus* are likely to proliferate several months after radiotherapy, thanks to a favorable environment created by xerostomia.

Responses to the reviewers' comments (Round 2):

Reviewer #2 (Comments for the Author):

no further comments; well done

Our response:

Thank you for your feedback; Cheers!

Reviewer #3 (Comments for the Author):

The authors have made corrections and provided clarifications on several points. However, the article does not sufficiently discuss the effects on the oral microbiome of the different treatments for OSCC. Many parameters, such as the host's diet, side effects of surgery and radiotherapy, and the administration of antibiotics can unbalance the microbiome. Similarly, local consequences such as the development of oral mucositis and dry mouth can also disrupt the microbial balance.

Most early or recurrent squamous cell carcinomas are treated with a surgical and radiotherapeutic approach, while, when implemented, chemotherapy (in the form of a platinum-based drug), allows radiosensitization of the cancer cells in the oral cavity.

Radiotherapy leads to an increase in the number of cariogenic bacteria such as *Streptococcus mutans* and *Lactobacillus*, as well as *Staphylococcus*, *Enterococcus* and the fungus *Candida albicans*, on the other hand there is a reduction in commensal microorganisms such as *Neisseria*. Species such as *Lactobacillus* are likely to proliferate several months after radiotherapy, thanks to a favorable environment created by xerostomia.

Our response:

We have added a relevant section in the manuscript to address this point and included the corresponding references:

“Although this study primarily focuses on the tumor microbiome characterization and OSCC diagnosis, the effects of therapeutic treatments on the oral microbiome require further investigation. For instance, standard treatments like surgery and radiotherapy, along with platinum-based chemotherapy for cancer cell radiosensitization, can substantially alter the oral microbiome (33, 54). Studies have shown that radiotherapy increases the abundance of cariogenic bacteria such as *Streptococcus mutans* and *Lactobacillus*, as well as *Staphylococcus*, *Enterococcus*, and the opportunistic pathogen *Candida albicans*, while reducing commensal bacteria like *Neisseria* (55). These microbial shifts are further complicated by treatment side effects like xerostomia, which creates conditions favorable for *Lactobacillus* proliferation that persist for months after radiotherapy (33). Therefore, additional multi-center studies with diverse patient populations are emergently needed to better understand the interactions between OSCC treatments and the oral microbiome.” (line 398-409)

Reference:

[33] Sami A, Elimairi I, Stanton C, Ross RP, Ryan CA. 2020. The Role of the Microbiome in Oral Squamous Cell Carcinoma with Insight into the Microbiome-Treatment Axis. *Int J Mol Sci* 21.

[54] Mougeot JC, Stevens CB, Almon KG, Paster BJ, Lalla RV, Brennan MT, Mougeot FB. 2019.

Caries-associated oral microbiome in head and neck cancer radiation patients: a longitudinal study. *J Oral Microbiol* 11:1586421.

[55] Almstahl A, Wikstrom M, Fagerberg-Mohlin B. 2008. Microflora in oral ecosystems in subjects with radiation-induced hyposalivation. *Oral Dis* 14:541-9.

Re: Spectrum00852-24R2 (Exploring oral microbiome in oral squamous cell carcinoma across environment-associated sample types)

Dear Prof. Haopeng Yu:

Your manuscript has been accepted, and I am forwarding it to the ASM production staff for publication. Your paper will first be checked to make sure all elements meet the technical requirements. ASM staff will contact you if anything needs to be revised before copyediting and production can begin. Otherwise, you will be notified when your proofs are ready to be viewed.

Sincerely,
Zhenjiang Xu
Editor
Microbiology Spectrum